# Diversity in Software Design and Construction Teaching: A Systematic Literature Review

**Vitor de Souza Castro** * and **Sandro Ronaldo Bezerra Oliveira** *

Graduate Program in Computer Science (PPGCC), Federal University of Pará, Belém 66075-110, PA, Brazil
* Correspondence: vitor@unifesspa.edu.br (V.d.S.C.); srbo@ufpa.br (S.R.B.O.)

**Abstract:** Teaching in computing faces challenges concerning technological changes and strategies to encourage student engagement with the teaching–learning process. In software engineering, specifically in the construction of the solution, these challenges are even greater due to technological changes and the evolution of applications. Based on this context, the objective of this work is to present the results of a systematic literature review on the knowledge areas that are central to the software construction process, the Software Design and Construction. The methodology for developing the systematic literature review followed a review protocol in which 51 studies were selected out of 302 studies found by executing a search string in the Association for Computing Machinery and Institute of Electrical and Electronic Engineers databases. As a main result, the diversity of teaching strategies applied to the teaching of Software Design and Construction was identified. In addition, amongst the selected studies, we identified that active methodologies are more frequent in the literature.

**Keywords:** teaching practices; software design and construction; systematic literature review

## 1. Introduction

Teaching in computing faces challenges concerning technological changes and strategies to encourage student engagement with the teaching–learning process. One of the pillars of computing is software development, its content providing students with knowledge and practices for the construction of quality software. In this context, Software Engineering serves as a pillar for the construction of quality software [1]. Bourque and Fairley [2] defined the areas of knowledge in the construction of software, among which are Software Design and Software Construction.

Higher education courses in Computer Science, Software Engineering, Information Systems and Computer Engineering present content and practices in the areas of software construction at different levels. According to Force [3], Computer Science and Software Engineering courses have more content related to software development than other Computer courses.

Teachers of software construction have a responsibility to make the teaching–learning process efficient, so as to enable the students' training to be successful. In this context, identifying teaching strategies, tools, and forms of alignment with the course are essential elements to ensure efficiency in the process.

The use of differentiated teaching–learning strategies enhances students' motivation and understanding of certain content [4,5]. In addition, making the student the protagonist of this process can be decisive for learning [6].

Understanding how, and which, teaching strategies can be applied to the teaching of computing, specifically in the software design and construction process, was the main motivation for the development of this work. Furthermore, the objective was also to understand how curricula are structured to support these teaching strategies.

Based on the context presented, this work aimed to present the results of the Systematic Literature Review (SLR) on the knowledge areas central to the process of constructing software, namely, Software Design and Construction, hereafter referred to as SDC. To this end, a review protocol was developed covering the fundamental research questions to identify the scenario that involves the teaching of SDC, detailed in Section 3. The SLR results present the context of teaching SDC of 51 selected studies and the diversity of strategies, forms of evaluation and tools used.

In addition to this introductory section, this article is organized as follows. Section 2 presents the theoretical foundation of this work. Section 3 presents the methodology, research questions and the SLR protocol used. In Section 4, the results obtained by executing the review protocol are presented. Section 5 presents reflections and discussions on the results. Section 6 presents threats to the validity of the study, and, finally, Section 7 presents final considerations, limitations and future work.

## 2. Background

### 2.1. Teaching in Software Engineering

Software Engineering (SE) is a subject that is part of Bachelor's degrees in Computer Science, Software Engineering, Information Systems and Computer Engineering. In the Pedagogical Projects of these courses, Software Engineering is a subject that aims to approach the main concepts of software conception, elaboration and development. According to Pressman, in [1], Software Engineering is about applying a systematic, disciplined and quantifiable approach to software development, operation and maintenance. In this sense, the emphasis of SE teaching covers the learning process associated with methodologies, tools and processes for software construction.

Sommerville, in [7], defined SE as an engineering subject which focuses on all aspects of software production, from the requirements for a design to its maintenance. For de Pádua Paula Filho [8], software engineering is understood as being complex, bringing together art, the meeting of human needs, scientific knowledge, empirical knowledge, specific skills, natural resources, and appropriate forms, devices, structures and processes.

Based on the concepts of software engineering, the complexity and diversity of content which needs to be taught to future professionals in computing courses are core issues. In the context of teaching software engineering, the exposition of theoretical content on methodologies and processes applied to software development is key. Lemos, Cunha and Saraiva, in [9], presented important elements about higher education, including the excessive focus on theory and content, rather than competence. In addition, it was reinforced that, in the Information Systems course, which was the object of study of their work, there is an inadequate sequence of subjects, which is one of the main influences in teaching–learning in SE.

In this sense, the present SLR seeks to identify, in the specialized literature, which strategies are used to enhance the teaching–learning of students, and, also, how course projects are organized for the teaching of SDC.

### 2.2. Product Design and Construction

For Sommerville [7], software design and construction is a stage in the process in which an executable software system is developed. At this stage of development, it is necessary that the requirements are validated, so that the design stage can begin. The software design stage encompasses the set of principles, concepts and practices that lead to the development of a product [1].

De Pádua Paula Filho, in [8], presents the design phase as being responsible for defining an implementable structure for a software product, which meets the specified requirements. For the construction phase, a system is designed in terms of different types of source code components and binary code, according to the chosen technologies. By definition, design and construction activities are related, with the objective of delivering a software product.

Within Software Engineering, design and construction are part of the software development process, which is related to software engineering support processes, such as Project Management and Configuration Management.

In software quality models, such as Capability Maturity Model Integration (CMMI), the software design and construction areas are included in the Technical Solution (TS) process area [10]. For the Brazilian Software Process Improvement (MPS.BR) model, the Product Design and Construction Process (PCP) has the purpose of designing, developing and implementing solutions to meet requirements [11].

Among the main characteristics of the Product Design and Construction process is the modeling of the software, based on the requirements and the implementation of the product, which refers to the coding of the software, based on the models developed in the design phase. Sommerville [7] states that Design and Construction are closely linked; the design must take into account the implementation issues. For the purposes of SLR, we consider the area related to Product Design and Implementation as Software Design and Construction (SDC).

### 2.3. Teaching and Learning Practices

Teaching and learning practices are presented as ways of organizing and exposing the content necessary to learn about a given subject. According to Glasser [12], learning can be achieved through different strategies, with reading being responsible for 10% of learning, and teaching responsible for 95% of learning, as shown in Figure 1.

In Figure 1, at the base of the pyramid, Teaching others and Practice represent the most efficient practices for learning, while practices that approach the top, such as Read and Listen, have lower learning efficiencies compared to those at the base.

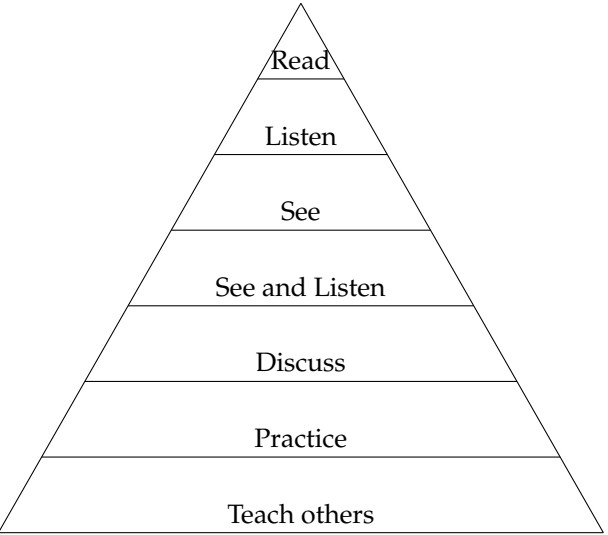

**Figure 1.** Adapted Glasser's Pyramid [12].

To enhance learning, active methodologies are presented as an alternative to facilitate the teaching–learning process. According to Berbel [13], active methodologies are based on ways of developing the learning process using real or simulated experiences, aimed at successfully solving challenges arising from the essential activities of social practice in different contexts. Among the most widely used active teaching practices are the Flipped Classroom and Project-Based Learning.

In this SLR, alternatives were sought that are actively aligned with the teaching–learning process. Identifying the impact of the application of these learning practices in the context of teaching SDC is necessary.

*2.4. Related Works*

This subsection aims to present some studies related to, directly or indirectly, the research. The scope of the research for the development of this section involved work on teaching strategies in computing courses, with a focus on SDC processes and related areas, such as Requirements Engineering.

A literature review for teaching programming was presented in [14]. The scope of the SLR was directed towards the programming of teaching–learning approaches in Brazil. The percentage of studies included was 3.13%, with 2325 studies listed and 73 included. The study by de Holanda and Freire [15] has similarities to [14], because it deals with an SLR to teaching programming; however, its scope was related to introductory programming, with 2.4% inclusion of studies from a total of 2109. Unlike [14,15], the work presented here carried out research to find strategies for teaching SDC without restricting the location of the study.

Curcio et al. [16] presented a systematic mapping of Requirements Engineering. The research selected 104 studies out of 2171; that is, 4.79% inclusion. The methodology for developing the systematic mapping made use of Snowballing to include three studies and the others were obtained through the execution of a search string in knowledge bases. The SLR presented here does not use Snowballing as a search strategy. In addition, the mapping is directed to the context of agile practices in Requirements Engineering.

In da Costa et al. [17] a systematic review of the literature on collaborative teaching strategies was carried out. In counterpoint, this work presents the teaching strategies in the specific context of Software Engineering. Besides this, the percentage of selected studies is 0.84%, with five selected studies out of 591 initial studies, making use of the ACM and IEEExplore knowledge bases.

Salleh et al. [18] presented an SLR on the use of Pair Programming in CS and SE courses. The objective of the SLR was to evaluate the efficiency and compatibility of Pair Programming in higher education. In Salleh et al. [18] the direction was to investigate agile practice and how it is presented in the scenario of higher education. For the work presented here, however, the focus is on understanding SDC teaching strategies, including peer learning.

Thus, the research presented in this article differs in the orientation of approach applied to teaching SDC. In addition, another relevant aspect is the information extracted from the studies, which selects the teaching units/courses, teaching strategies, tools used, and the arrangement of groups of students in the projects; a scenario not observed in the identified related studies.

## 3. Materials and Methods

*3.1. Objectives and Research Questions*

The research presented here aims to identify strategies applied to the teaching of software engineering, and which are directed to content related to Software Design and Construction in computing courses (Computer Science, Information Systems and Software Engineering). There is a hypothesis that, in the pedagogical project of the course, the focus of the subject highlights the teaching of software methods and processes, prioritizing the area of requirements engineering.

To develop the research questions, the PICOC strategy was used, which provides useful tools to define clear and focused questions and to develop a review protocol, with the acronym (P—Population, I—Intervention, C—Comparison, O—Outcome and C—Context) [19,20]. Table 1 presents the parameters used to develop the research questions of this work.

Given the hypothesis presented, and application of the PICOC strategy, as shown in Table 1, we proposed the following research questions:

Q1. How do the computer courses approach the knowledge area of Software Design and Construction?

Q2. How does the teaching of Software Engineering present the contents related to the knowledge area of Software Design and Construction?

Q3. What strategies (methods, techniques, tools, approaches) are used for teaching software design and construction in the context of Software Engineering?

Q4. How were the strategies (obtained as results in Q3) used for teaching Software Design and Construction in the context of Software Engineering evaluated?

**Table 1.** PICOC parameters.

| Parameter | Description |
| --- | --- |
| **P**opulation | undergraduate courses in computing |
| **I**ntervention | teaching methods, techniques, tools and practices |
| **C**omparison | not applicable |
| **O**utcome | strategies for teaching Software Design and Construction |
| **C**ontext | academic studies |

### 3.2. Methodology

The objective of this work was to research and investigate aspects related to the teaching and learning process of SDC through a Systematic Literature Review (SLR). An SLR is a means of identifying, evaluating and interpreting all available research which may be relevant to a particular research question, topic area or phenomenon of interest [21].

To this end, the following steps were used in this work, based on the studies by [17,21,22]:

1. Review Planning

    (a) Identify the research question(s),
    (b) Search strategy, search terms and search strings,
    (c) Define criteria for Inclusion and exclusion to the primary studies,
    (d) Define criteria for classification of studies,
    (e) Define dataset for extraction in studies.

2. Review Process

    (a) Identify and extract data from relevant studies,
    (b) Apply the selection criteria established in phase 1,
    (c) Synthesize the obtained data.

3. Review Documentation:

    (a) Develop, validate and present a review report.

### 3.3. Research Strategy

The knowledge repositories selected for the research were the ACM Digital Library and IEEE Xplore Digital Library. Both libraries provide full access to content and the possibility to search the full text.

Based on the research objectives, Section 3.1, we developed the following search string:

("curriculum" OR "discipline" OR "course" OR "subject")

AND ("graduate" OR "undergraduate" OR "computer science" OR "information systems")

AND ("software engineering")

AND ("technical soluction" OR "design" OR "solution" OR "implementation" OR "construction" OR "integration" OR "architecture" OR "software" OR "component" OR "interface" OR "connection" OR "product").

Regarding the time period of the studies, we chose the last 8 years (from 2015 to 2022), because, in 2015, the Curricular Guidelines for the undergraduate course in Software Engineering, by ACM/IEEE, were published, which, in 2022, were updated. As for the guidelines for Information Systems and Computer Science courses, recent updates were in

the years 2021 and 2020, respectively. In addition, the Brazilian Computer Society (SBC) curriculum guidelines for computing courses was updated in 2017.

Considering these time frames for updating the curricular guidelines by the ACM/IEEE and SBC, the time period of the studies researched was from 2015 to 2022, and aimed to cover studies published from 2015 onwards.

The search string performed on the databases was limited to the Title of the Work and Abstract fields, with the objective of obtaining greater relevance in the studies. For validation of the search string, studies by [23–26], that met the research questions, were selected and it was found, in the process of executing the search string in the selected databases, that the studies were listed.

### 3.4. Selection of Studies

For the selection of studies, inclusion criteria (IC) and exclusion criteria (EC) were adopted in order to select relevant studies that aligned with the objectives and research questions, as follows:

Inclusion Criteria

1.  studies that presented some strategy applied to the teaching of computing, were in the classroom, and directed to the knowledge area of Software Design and Construction.

Exclusion Criteria

1.  studies in languages other than English and Portuguese,
2.  studies that were not in full format [27],
3.  studies with access restriction,
4.  duplicate studies.

The selection of studies was carried out in phases with the aim of applying the exclusion criteria in order to obtain the relevant studies. Table 2 presents the selection phases applied to the researched studies, according to Section 3.3.

**Table 2.** Study Selection Phases.

| Phase | Activity |
| --- | --- |
| I | Elimination by applying the established exclusion criteria |
| II | Elimination by Title and Abstract by application of the established inclusion criteria |
| III | Data extraction and classification by Full reading |

The studies selected in phase I were submitted to phase II, according to the inclusion criteria, and any study that did not meet a criterion was eliminated. In phase III, data were extracted and the quality criteria, defined in Table 3, were applied in order to classify the studies.

### 3.5. Classification of Studies and Data Extraction

Kitchenham and Charters [21] consider the inclusion of quality criteria for the classification of selected studies as fundamental. In this sense, Table 3 presents the criteria applied in this review.

After the conclusion of phase III, according to Table 2, it was necessary to apply the quality criteria, established in Section 3.5, in order to classify the studies with greater relevance to the research.

For the evaluation of each criterion, defined in Table 3, the Likert agreement scale was utilized: Totally Agree (100% of points), Partially Agree (75% of Points), Partially Disagree (50 % of points) and Strongly Disagree (0% of points).

The data collected after the completed phases of studies comprised the following: title, author, year of publication and methods, tools and practices cited, which undergraduate

course the study applied to, which research evaluation methods were used, and which subject evaluation methods were adopted.

**Table 3.** Quality criteria applied.

| Criteria | Points |
|---|---|
| The study clearly defines the research objective (defines research question). | 1 |
| The study answers the defined research questions. | 1 |
| Methods, tools and practices for teaching software design and construction were mentioned in the study | 3 |
| References in the course curriculum regarding the theme of Software Design and Construction were indicated. | 3 |
| The evaluation of the application of the method, tools and/or practices for the teaching of Design and Construction of the software was defined and applied. | 2 |
| TOTAL | 10 |

### 3.6. Documentation and Presentation of Results Strategy

The review's documentation strategy made use of the Mendely Tool to store the studies in digital format, references and researchers' notes. Furthermore, the tool allowed the inclusion of tags in order to facilitate the identification of which phase of any given study was eliminated. In order to apply the quality criteria defined in Section 3.5, online electronic spreadsheets were used to release the grades by criterion and to order the scores.

Finally, the data were imported into a MySQL database in order to perform the queries and groupings of the research results that served as input for the presentation of the results. The presentation of the results was through tables and charts for quantitative analysis of the data generated by the research.

### 4. Results

*4.1. Overview of Phases*

As a result of executing the search string in the selected databases, Figure 2 shows the number of studies per year submitted to selection phases I, II and III, as defined in Table 2.

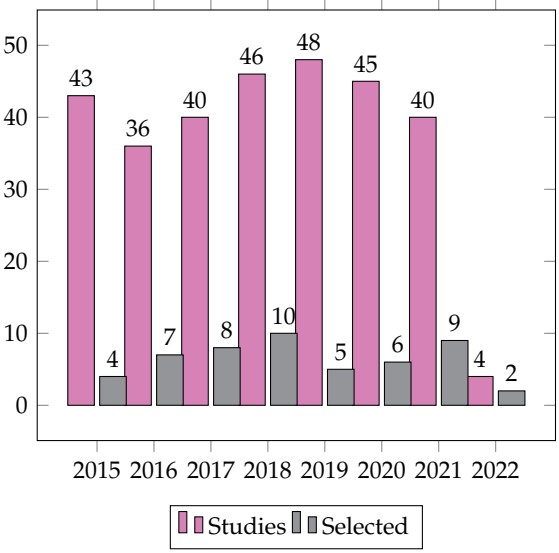

**Figure 2.** Number of studies per year.

Upon completion of the study selection phases, Table 4 presents the number of studies selected in each phase. It is noteworthy that the studies selected in Phase III, used for data extraction in order to obtain answers to the research questions, totaled 51 studies, representing 16.88% of the total studies searched, as can be seen in Appendix A.

**Table 4.** Execution of Phases

| Repositories | Studies | Phase I | Phase II | Phase III |
|---|---|---|---|---|
| ACM | 111 | 29 | 57 | 25 |
| IEEE | 191 | 63 | 102 | 26 |
| Total | 302 | 92 | 159 | 51 |

As for the distribution of studies for each year, Figure 2, there is a similarity, in quantitative terms, of the number of articles searched and articles selected, except for the year 2022, as the execution of the research was held in the first quarter of that year.

As a mechanism for classifying the studies, the evaluation criteria were applied to the selected studies. Figure 3 presents the percentages of works by point range (0 to 10), as established in Table 3. Considering that evaluations greater than 7 indicate studies with a higher quality indicator, it is noted that the research included 62.75% of selected studies. Only 11.76% of the studies received an evaluation of less than 5 points. In contrast, only studies PS03, PS10, PS14, PS16 and PS18 received a score of 10, fully meeting all the quality criteria.

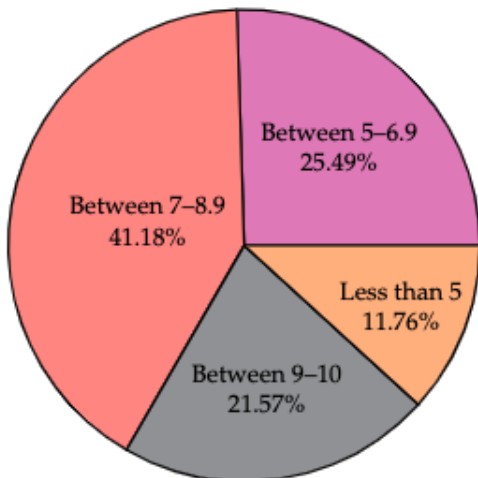

**Figure 3.** Quality Evaluation.

In regard to the undergraduate courses with the highest incidence of selected studies, the following 4 courses were found to have the greatest representation among the 15 different courses: Computer Science (CS) with 50.98%, Software Engineering (SE) with 31.37%, Computer Engineering (CE) with 3.92% and Information Systems (IS) with 3.92%. Among the 15 courses found, PS13, PS29, PS34 and PS49 studies were from Postgraduate courses in Software Engineering.

The higher incidence of the Computer Science course is due to the fact the course emphasizes the teaching of Software Design and Construction (SDC), reflected in the areas of Fundamentals of Software Development, Fundamentals of Systems and Software Engineering totaling 40.60% of the course load [28].

The CS and SE courses predominated in the studies due to their curricular organization and course emphasis on Software Development, Systems Infrastructure and Technological Applications, in contrast to the IS course, which focuses on Systems Organization [3] (p. 39).

Research evaluation methods were among the data extracted from the selected studies, and Table 5 presents the evaluation methods, the studies that used the methods and the percentage of use of the methods among the selected studies.

**Table 5.** Research evaluation method.

| Evaluation Method | Studies | Percentage |
|---|---|---|
| Questionnaire | PS04 PS05 PS07 PS08 PS09 PS10 PS11 PS12 PS13 PS14 PS15 PS16 PS18 PS19 PS20 PS21 PS22 PS23 PS25 PS27 PS28 PS29 PS32 PS35 PS36 PS37 PS39 PS40 PS41 PS42 PS44 PS45 PS47 PS48 PS50 | 53.85% |
| Lessons Learned Record | PS03 PS17 PS34 PS36 PS38 PS43 | 9.23% |
| Statistical Analysis | PS06 PS11 PS18 PS19 PS30 | 7.69% |
| Note | PS05 PS24 PS40 PS51 | 6.15% |
| Correlation Analysis | PS16 PS23 PS29 | 4.62% |
| Interview | PS02 PS21 | 3.08% |
| Not Shown | PS31 PS46 | 3.08% |
| Experience report | PS33 PS47 | 3.08% |
| Data Analysis | PS40 | 1.54% |
| Tool-generated data analysis | PS26 | 1.54% |
| Documentation Review | PS49 | 1.54% |
| Code Quality Rating | PS12 | 1.54% |
| Evaluation Events | PS01 | 1.54% |
| Peer review | PS04 | 1.54% |

It is noted that the use of a Questionnaire was the method used in 53.85% of the studies, being the main tool to obtain structured data, through multiple-choice questions, and to obtain an evaluation through comments made by the participants of the research using open-ended questions.

Among the methods used, it can also be highlighted that the analysis of data generated by tools and the evaluation of code quality were unique methods used. In PS26, the tool used the data from the commitment made by students to present the evaluation on compiling the source code, executing the unit tests and performing static analyses on the submitted code. As a result of this work, there was a stimulus for the development of the project and the number of activities submitted before the deadline, encouraging students to point out indicators to improve the source code produced.

In PS12, the objective was to submit the source code produced by the students on a website for quality evaluation, with the result generated by the evaluation website being a mandatory item for the delivery of the activity. The results of this work indicated that the execution of code evaluation as an evaluation item enabled the improvement of the students' code, and 70% of the survey respondents identified benefits in the use of the teaching method.

To organize the information, Sections 4.2–4.5 present the results obtained for each research question.

### 4.2. Q1—How Do the Computer Courses Approach the Knowledge Area of Software Design and Construction (SDC)?

Research question 1, Q1, sought to identify how the computer courses presented the SDC knowledge area; in short, in which curricular structures SDC content was present. In the data extraction process to answer Q1, it was identified that Teaching Unit or Subjects were related to the teaching of SDC.

The five Teaching Units/Subjects in which the SDC knowledge area was addressed with the highest percentage among the selected studies were:

1. Software Engineering (SE): PS01, PS03, PS06, PS13, PS14, PS19, PS20, PS21, PS22, PS24, PS26, PS36, PS42, PS43, PS44, PS45 and PS47 studies, with 20.24%,
2. Software Architecture (SA): PS15, PS39 and PS50 studies, with 3.57%,
3. Database (DB): PS01 and PS03 studies, with 2.38%,
4. Data Structure (DS): PS05 and PS26 studies, with 2.38%, and
5. Software Engineering I (SE-I): PS05 and PS23 studies, with 2.38%.

It is noteworthy that among the selected studies there were 56 different Teaching Units/Subjects. However, only those mentioned in the graph in Figure 5 had more than one occurrence. In addition, 7 studies (PS02, PS28, PS29, PS30, PS32, PS46 and PS49) did not mention the teaching unit/subject, as they were general studies about the course, internships, extension programs or analytical studies about the syllabus.

Based on the results obtained, PS11 stood out. It presented the implementation of the teaching unit called SE-First, with the objective of integrating non-technical skills (soft skills) to  Software Engineering before students effectively participated in subjects with technical characteristics.  As a result of this study, cited in the work, students who participated in SE-First showed greater learning than students who participated in traditional approaches without using SE-First.

From the results, it is noted that Software Engineering was the main teaching unit to present SDC content.

### 4.3. Q2 — How Does the Teaching of Software Engineering Present the Content Related to the Knowledge Area of Software Design and Construction?

Research question 2, Q2, sought to identify how SDC teaching is presented, and, for this purpose, the following classification was used:

- Theoretical: presentation of the content of the teaching unit/subject in an expositive way, aimed at conceptualization,
- Practical: carrying out activities that involve the application of the content in a practical way, through the development of projects, and practice at problem solving in laboratories,
- Theoretical and Practical: work that uses both approaches together.

Figure 4 shows 54.90% for teaching SDC in a practical way and 35.29% for theoretical and practical, which reinforces the fact that teaching of SDC is mostly presented in a practical way, compared to a theoretical approach.

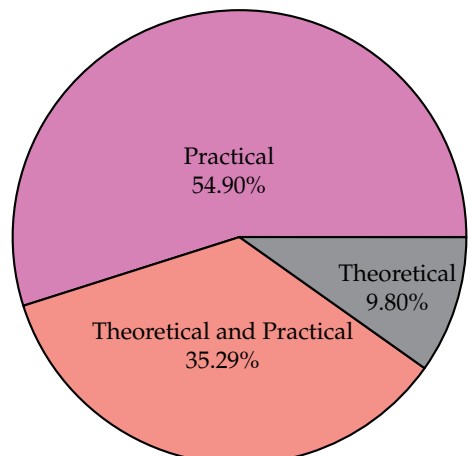

**Figure 4.** Teaching model.

In addition, research question Q2 extracted information on the involvement of students in the teaching–learning process, either actively, when the student is the protagonist and autonomous in the teaching–learning process, or passively, when the student is a receiver of information passed on by the teacher without active participation in this process.

In the selected studies, from the perspective of the teaching approach being active or passive, 90.20% of the studies were found to be active, and 9.80% passive. This is mainly due to the use of practical activities, many of them conducted in groups, which encourage students to play a leading role in the teaching–learning process.

Among the studies with exclusively practical teaching approaches, the courses with the highest percentage were CS, with 40.63%, and SE, with 28.13%, and the other courses

had 3.13%. It is noted that the CS and SE courses had a higher incidence of practical work due to the curricular structure being focused on Software Development, Systems Fundamentals and Software Engineering activities [28].

*4.4. Q3—What Strategies (Methods, Techniques, Tools, Approaches) Are Used for Teaching Software Design and Construction in the Context of Software Engineering?*

Research question 3, Q3, aimed to identify which strategies were used for teaching SDC. The methods, techniques and approaches used to present content, whether in a practical or theoretical way, were considered as strategies for teaching SDC.

Due to the identification of strategies involving group work among the selected studies, this subsection presents the number of members of the groups and the type of stakeholders when an activity involved software project development.

Table 6 presents the identified strategies, as well as which studies employed the strategies and the percentage of occurrence among the selected studies. It is noted that the formation of a group for project development was the most used strategy. In addition, several practices were found, which highlights the diversity of ways of teaching SDC.

**Table 6.** Strategies used for teaching Software Design and Construction.

| Strategies | Studies | Percentage |
|---|---|---|
| Project Development Group | PS28 PS45 PS34 PS32 PS36 PS31 PS30 PS37 PS16 PS44 PS27 PS10 PS25 PS48 PS24 PS50 PS38 PS21 PS43 | 22.35% |
| Project Based Learning | PS19 PS23 PS14 PS40 PS13 PS02 PS08 PS03 PS07 PS49 PS05 PS04 | 14.12% |
| Lab in hands-on format | PS20 PS44 PS33 PS31 PS11 PS17 PS50 | 8.24% |
| Problem Based Learning | PS29 PS16 PS35 PS49 PS13 | 5.88% |
| Flipped Classroom | PS34 PS06 PS23 PS05 PS18 | 5.88% |
| Practical Classes | PS18 PS07 PS46 PS36 | 4.71% |
| Theoretical Classes | PS09 PS36 PS07 PS48 | 4.71% |
| Educational Games | PS13 PS15 PS42 | 3.53% |
| Use of Design Thinking | PS37 PS24 | 2.35% |
| Seminars | PS14 PS20 | 2.35% |
| Formal Methods | PS22 | 1.18% |
| Workshops | PS29 | 1.18% |
| Video Production | PS14 | 1.18% |
| Pair Programming | PS03 | 1.18% |
| Service Learning Project | PS01 | 1.18% |
| Quizz | PS13 | 1.18% |
| Use of Comparative Critical Tool | PS39 | 1.18% |
| Use of tool for automating analysis and evaluation | PS26 | 1.18% |
| Use of Kahoot | PS18 | 1.18% |
| Athletic Approach | PS47 | 1.18% |
| Collaborative Learning Method | PS41 | 1.18% |
| Mind Maps | PS13 | 1.18% |
| Instruction by pairs | PS06 | 1.18% |
| Design Reflection Framework | PS51 | 1.18% |
| Formative Feedback | PS48 | 1.18% |
| Students Generating Questions | PS08 | 1.18% |
| Design Challenges | PS51 | 1.18% |
| Peer Learning | PS16 | 1.18% |
| Inquiry Based Learning | PS16 | 1.18% |
| Example Based Learning | PS39 | 1.18% |
| Crowd-based Learning | PS12 | 1.18% |
| Case Based Learning | PS29 | 1.18% |

Regarding the strategies, presented in Table 6, the following active learning methods were identified as standing out: Project-Based Learning, Problem-Based Learning and Flipped Classroom.

PS47 was about an athletic teaching approach that was applied in 3 subjects in the Software Engineering course. This practice seeks to develop the student's skills with a focus on efficiency and performance. The execution of the approach consists of presenting a problem to the students and a video demonstrating an alternative solution in the optimal resolution time. The video allows students to understand the importance of finding a solution to apply to a specific problem within time. Based on the problem, the teacher decides the necessary time interval to solve it, basing the maximum time for the activity on the optimal time. Students develop the solution to the problem presented in a timely way in the classroom, known as Workout. If the student exceeds the maximum time, he or she must review the solution and restart the whole process, resetting the timer to build the solution again.

The results of the work in PS47 indicated that 96% of the research participants preferred the use of an athletic approach, compared to the traditional teaching approach, and, in addition, 82% of the research participants indicated that Workout brought focus to development of a problem.

In addition, another study that represents the diversity of approaches to teaching SDC is the work in PS41, in which the focus was the use of the collaborative learning method in requirements engineering for software diagram evaluation by students. The method consists of dividing students into teams to discuss a topic. The teacher divides the topic into parts and the team decides which student takes notes on each part. After that, students from different teams with the same part of the topic group together to share information about the study. Finally, students return to their original groups to share the results of the discussions and to develop a presentation on the specific topic. As a result, 75% of the students who participated in the survey demonstrated satisfaction with the use of collaborative participation and highlighted a better understanding of software modeling content.

The review also extracted the amount of each strategy, in percentage, used to teach SDC, as shown in Figure 7. It is noted that the use of one strategy accounted for 54.90% and, in the two strategies, it represented 29.41% of the studies. There were studies, representing 15.69%, that used more than two strategies to teach SDC.

PS13, for instance, combined 5 strategies: Problem-Based Learning, Project-Based Learning, Educational Games, Mind Maps and Quizzes. As a result, the research participants indicated that the integration of active methodologies with gamification increased student engagement in the classroom. Unlike PS13, PS16 presented the combination of 4 strategies restricted to the following active methodologies: Problem-Based Learning, Investigation-Based Learning, Peer Learning and Project Development Group. The results of the studies identified that the Problem-Based Learning strategy was the most efficient for learning, compared to the others used in the study. It was identified that students who participated in the study using such strategies obtained better grades than groups that did not use them.

In regard to the studies that presented the development of SDC activities as teamwork, the number of participants in each team was taken into consideration. The grouping carried out to generate Figure 5 follows: maximum 4 members (2, 2 to 3, 2 to 4 and 3 to 4), maximum 6 members (3 to 5, 3 to 6, 4 to 5, 4 to 6, 5 and 5 to 6) and a maximum of 8 members (6 to 8 and 8).

Regarding team activities, information was extracted about the origin of the project's stakeholder, and the works were classified as follows:

- Not defined: No indications of project stakeholder identity. In some cases, the team itself assumed this role, such as PS04, PS05, PS07, PS10, PS12, PS13, PS14, PS16, PS20, PS23, PS25, PS31, PS37, PS45, PS46 and PS50,
- Internal: Subject teacher or monitor characterized as stakeholder, as in PS19, PS24, PS36 and PS40,

- External: Person external to the subject characterized as stakeholder. In many cases, this was a representative of a company for which the group was developing a project, as in PS01, PS02, PS03, PS28, PS30, PS32, PS38, PS43 and PS44.

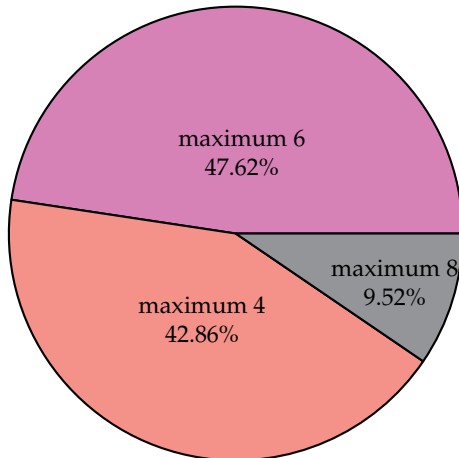

**Figure 5.** Team Size.

In this regard, 43.13% of the studies did not have strategies that required stakeholders for the projects. Figure 6 presents the distribution by classification among the works that required a stakeholder role. Analysis of this indicator, and removal of works classified as "Not defined", exhibited that 69.23% of the works had external stakeholders.

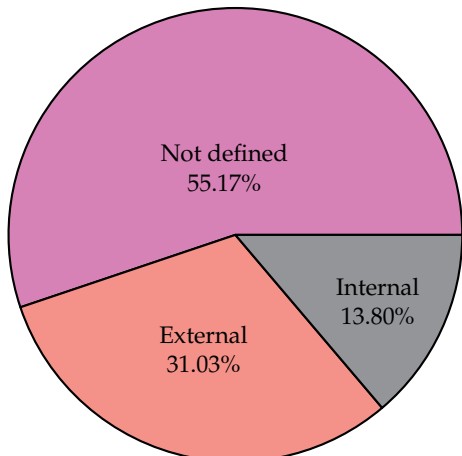

**Figure 6.** Stakeholder type.

Among the works that made use of external stakeholders, PS02 stood out. In this study, the Project-Based Learning teaching strategy was used in the software engineering course so as to present how the strategy stimulated the collaboration of students with stakeholders, from the perspective of seeking solutions for a given problem. Each student had a specific role within the project and there was a different stakeholder for each group. Despite the projects being developed for application in a real environment, the stakeholders were not interested in the finished product, but rather in ideas to solve the requisite problem. As a result, through interviews, 90% of stakeholders recommended the use of this teaching strategy.

Regarding the way students were selected for the teams, only PS13, PS14, PS34 and PS45 presented the format used. In all, 75% of the studies used automated tools, taking into account answers to a questionnaire, to organize teams.

In PS34 and PS45, the tool chosen to divide the teams was CATME (Available online: https://info.catme.org/ (accessed on 20 December 2022)). In PS45, personal criteria were

included for team formation, such as the following: gender, average grade, available time for the agenda and level of commitment. In addition, the tool was used to carry out peer evaluation among students on the same team for the following items: contribution to the team, interaction with the team, knowledge and skills.

In the case of tools and technologies used in teaching SDC, 50% of the studies did not present tools or technologies, according to Table 7. This is due to the emphasis of the works on the evaluation of teaching strategy and not of the technology or tool used for teaching SDC.

Another aspect worth highlighting is the use of tools that operate in SDC support processes, such as Project Management and Software Configuration Management. The use of Git as the project's version control technology (Git, Github and Gitlab) wss present in 10.18% of the studies. Tools directed towards Project Management (Trello, MS Project, Taiga, Zenhub and Agilefant) were in 3.72% of the studies, which demonstrates the possibility of combining other areas of Software Engineering with teaching of SDC.

**Table 7.** Tools used.

| Tool | Studies | Percentage |
|---|---|---|
| Not presented | PS01 PS02 PS08 PS11 PS19 PS25 PS27 PS28 PS29 PS31 PS32 PS33 PS35 PS36 PS38 PS41 PS43 PS44 PS47 PS48 PS49 PS50 PS51 | 21.30% |
| Github | PS07 PS12 PS14 PS17 PS20 PS21 PS26 | 6.48% |
| UML | PS04 PS09 PS21 PS22 PS23 PS24 | 5.56% |
| SonarQube | PS10 PS20 PS30 | 2.78% |
| JAVA | PS09 PS17 PS46 | 2.78% |
| Heroku | PS40 PS46 | 1.85% |
| Slack | PS14 PS40 | 1.85% |
| Gitlab | PS10 PS30 | 1.85% |
| Git | PS34 PS46 | 1.85% |
| Eclipse | PS13 PS26 | 1.85% |
| JUnit | PS09 PS13 | 1.85% |
| CATME | PS34 PS45 | 1.85% |
| Astah | PS03 PS13 | 1.85% |
| Se-RPG | PS13 | 0.93% |
| ScreenFlow | PS34 | 0.93% |
| Postgres | PS46 | 0.93% |
| Selenium | PS30 | 0.93% |
| Panopto | PS34 | 0.93% |
| OpenShift | PS40 | 0.93% |
| Skype | PS40 | 0.93% |
| node.js | PS34 | 0.93% |
| Socrative | PS16 | 0.93% |
| StarUML | PS21 | 0.93% |
| SoftBook | PS42 | 0.93% |
| StackOverFlow | PS12 | 0.93% |
| LEARN Board Game | PS15 | 0.93% |
| Taiga | PS40 | 0.93% |
| Teamspeak | PS40 | 0.93% |
| Coverage Test | PS13 | 0.93% |
| Trello | PS40 | 0.93% |
| Uppaal | PS22 | 0.93% |
| uTest | PS13 | 0.93% |
| XML | PS09 | 0.93% |
| xUnit | PS30 | 0.93% |
| Zenhub | PS14 | 0.93% |
| Zoom | PS06 | 0.93% |
| Google Drive | PS05 | 0.93% |
| ArgoUML | PS21 | 0.93% |

**Table 7.** *Cont.*

| Tool | Studies | Percentage |
|---|---|---|
| Blog | PS37 | 0.93% |
| Bluemix | PS40 | 0.93% |
| Canary Framework | PS26 | 0.93% |
| Codeface | PS30 | 0.93% |
| CSDCT (client-server design critic tool) | PS39 | 0.93% |
| Docker | PS30 | 0.93% |
| Draw.io | PS05 | 0.93% |
| Eclemma | PS13 | 0.93% |
| express.js | PS34 | 0.93% |
| FreeMind | PS13 | 0.93% |
| Gmail | PS05 | 0.93% |
| Google Classroom | PS21 | 0.93% |
| MySQL Workbench | PS03 | 0.93% |
| Gredos | PS05 | 0.93% |
| IdVSAL | PS05 | 0.93% |
| Island of Requirement | PS13 | 0.93% |
| IslandTest | PS13 | 0.93% |
| iTestLearning | PS13 | 0.93% |
| Jenkins | PS26 | 0.93% |
| Kahoot | PS18 | 0.93% |
| Agilefant | PS10 | 0.93% |
| Metric for Eclipse | PS13 | 0.93% |
| Moodle | PS05 | 0.93% |
| MS Project | PS20 | 0.93% |
| MySQL | PS09 | 0.93% |

*4.5. Q4—How Were the Strategies, (Obtained as Results to Q3), Used to Teach Software Design and Construction in the Context of Software Engineering Evaluated?*

Research Question 4, Q4, aimed to identify how strategies for teaching SDC were evaluated. The forms reflect what types of evaluation teachers use to measure the teaching–learning process in teaching units/subjects. Table 8 presents a summary of the strategies identified in the studies, as well as the percentage of incidence. It is noted that the highest percentage of works did not present an evaluation strategy, namely, 12.20%.

Question Q4 identified the diversity of forms of evaluation used in teaching SDC, with 49 strategies, among the selected studies.

It is worth mentioning that the evaluation strategies aimed at delivering projects, whether final or partial, obtained 9.76% and 4.88%, respectively. Regarding the diversity of evaluation, PS40 and PS48 stood out with 7 evaluation strategies applied together in the same study. Despite the use of different strategies to evaluate the teaching of SDC, the studies did not explore the impact of number of strategies applied to teaching.

PS40 presented 7 evaluation strategies: Oral presentation, Individual contribution to the project, Final delivery of the Project, Partial delivery of the Project, Laboratory, Test and Quizz. The objective of this work was to carry out the evaluation of the SE subject over 6 semesters of Project-Based Learning. In this study, agile practices, such as the following, were used in the development of projects by students: interactive and incremental development, collective ownership, fixed-size interactions, continuous delivery, periodic meeting, simple project and pair programming. Regarding teaching evaluations, individual project grades were defined for activities assigned in each project interaction in conjunction with the evaluation of the project repository by the professor and monitor. In addition, there were presentations of the projects by the teams, one partial and one final, and both with evaluation weight.

**Table 8.** Methods of evaluation used in teaching Software Design and Construction.

| Evaluation Method | Studies | Percentage |
| --- | --- | --- |
| Not shown | PS02 PS11 PS12 PS15 PS17 PS21 PS25 PS28 PS31 PS32 PS39 PS42 PS45 PS46 PS51 | 12.20% |
| Final Project Delivery | PS05 PS08 PS16 PS18 PS23 PS24 PS27 PS40 PS43 PS44 PS48 PS50 | 9.76% |
| Exercise | PS05 PS06 PS09 PS22 PS23 PS33 PS36 PS48 PS50 | 7.32% |
| Quizz | PS06 PS08 PS09 PS27 PS34 PS40 PS41 PS50 | 6.50% |
| Written test | PS06 PS09 PS16 PS18 PS33 PS48 PS49 | 5.69% |
| Project partial delivery | PS05 PS10 PS14 PS18 PS23 PS40 | 4.88% |
| Oral presentation | PS40 PS43 PS44 PS48 PS49 | 4.07% |
| Test | PS27 PS36 PS40 PS44 | 3.25% |
| Workshops | PS05 PS23 PS29 | 2.44% |
| Presence | PS05 PS37 PS38 | 2.44% |
| Oral presentation of the project | PS04 PS07 PS16 | 2.44% |
| Rating based on tool data | PS20 PS26 PS30 | 2.44% |
| Work Product Analysis | PS03 PS19 | 1.63% |
| Article production | PS13 PS50 | 1.63% |
| Practical Test | PS23 PS33 | 1.63% |
| Participation | PS05 PS34 | 1.63% |
| Lab | PS34 PS40 | 1.63% |
| Team Weekly Report | PS03 PS38 | 1.63% |
| Peer review of project members | PS10 PS44 | 1.63% |
| Individual Learning Summary | PS48 | 0.81% |
| Individual Activity Report | PS38 | 0.81% |
| Final report | PS04 | 0.81% |
| Project report | PS49 | 0.81% |
| Laboratory Report | PS27 | 0.81% |
| Peer review by students | PS33 | 0.81% |
| Final presentation of the project | PS34 | 0.81% |
| Theoretical Test | PS23 | 0.81% |
| Practical Test | PS16 | 0.81% |
| Laboratory Work | PS33 | 0.81% |
| Workout | PS47 | 0.81% |
| Oral test | PS49 | 0.81% |
| Final exam | PS06 | 0.81% |
| Project Video Production | PS14 | 0.81% |
| Evaluation Event | PS01 | 0.81% |
| Evaluation of problems solved individually | PS35 | 0.81% |
| Individual evaluation in the project | PS10 | 0.81% |
| Peer review by teachers | PS04 | 0.81% |
| Individual contribution to the project | PS40 | 0.81% |
| Project Performance | PS34 | 0.81% |
| Project Documentation | PS37 | 0.81% |
| Project rating by another team | PS48 | 0.81% |
| Project Blog Rating | PS37 | 0.81% |
| Gamification activities | PS13 | 0.81% |
| External contributor rating | PS48 | 0.81% |
| Reading Exercise | PS27 | 0.81% |
| Learning Memorial | PS38 | 0.81% |
| Handwriting evaluation of deliverables | PS29 | 0.81% |
| Project Poster | PS37 | 0.81% |
| Evaluation of work products | PS36 | 0.81% |
| Video Production | PS37 | 0.81% |

PS48 aimed to apply the constructivist alignment theory in two subjects with the use of formative feedback and late summative evaluation. Regarding the teaching evaluations, each week the students submitted the activities and the teacher performed the evaluation (without assigning a grade) and provided formative feedback. Students were encouraged to incorporate feedback and resubmit the activity. At the end of the teaching unit, the teacher evaluated the portfolio of activities sent by the student. In practice, the student submitted the Activity Portfolio at the end, including all the work products that helped in learning, such as notes, mental maps, reports, source code, etc.

Among the selected studies, only 8 presented the weights of each evaluation item for the final grade, with 6 having project development subjects, according to Table 9. Among the weights attributed to the project, a maximum percentage of 80% and minimum of 33% were indicated. These percentages contributed to the development of the teaching unit with a focus on practical activities, and, in particular, the development of projects. Regarding the percentage difference of the project's weight in the final evaluation, the diversity and non-standardization in the evaluation process for the teaching units that involved the development of projects was evident.

**Table 9.** Project weight in the final grade.

| Studies | Evaluation Components | Project Weight |
|---------|----------------------|----------------|
| PS05 | Project, Final Exam and Continuous Evaluation | 35% |
| PS18 | Project and Activities | 60% |
| PS27 | Project, Quiz, Lab Report and Tests | 36% |
| PS34 | Project, Quiz, Participation, Project Performance and Lab | 45% |
| PS37 | Project, Exercises, Test and Evaluation of Work Products | 33% |
| PS40 | Project, Lab, Test and Quiz | 80% |

## 5. Discussion

This section carries out a critical analysis of the selected studies, making use of the extracted data, based on the defined research questions, in order to understand the behavior of, and to indicate justifications for, such phenomena.

The studies presented in Section 4 demonstrate the diversity of strategies for teaching SDC. It is noted that the use of group strategies obtained the highest percentage among the selected studies. On the other hand, new approaches were identified, such as Formative Feedback (PS48) and the Athletic Approach (PS47), that, although restricted to one work each, demonstrated the possibility of using other techniques for teaching SDC.

The Problem-Based and Project-Based Learning Approaches had greater visibility and incidence in the literature, as they are active teaching strategies, which may involve the formation of groups among students to present a solution for a given case. In addition, they promote student experience in developing solutions, which involves holistic thinking about the content of the course and the use of soft-skills, skills that are often not emphasized in training courses in Information and Communication Technology.

Traditional teaching was present in this SLR through the use of theoretical classes, practical classes in laboratories and seminars. This indicates that, despite the diversity of practices for teaching SDC, traditional strategies are used in this context.

Regarding the modality types of SDC teaching, the use of Practical teaching had greater incidence in the selected studies. In addition, the active teaching approach was predominant among the works. A justification for this scenario is the use of teaching strategies with greater interaction among students, sharing the protagonist role of the teaching–learning process.

As for the evaluation methods of SDC teaching, diversity was considered. Traditional tests, whether written or practical, were present in the selected works. However, the use of Learning Memorial (PS38) and Video Production (PS14 and PS37) were presented as alternative ways to carry out evaluation of students.

Since the teaching of SDC involves activities with a greater need for practice, such as the development of a software project and its construction, the following elements of the evaluation method aimed at evaluating the project were evident: partial and final delivery of the project, work product analysis, project documentation, project report, and project video production, among others. Despite this diversity in evaluation methods, the emphasis of the selected works was not directed on the evaluation of the methods, but rather to teaching strategies.

Another important aspect was the fragmentation of SDC teaching into two or more teaching units/subjects. PS10, for instance, was only directed at project development, without the inclusion of theories to support the practice. On the other hand, PS04 addressed, in the same semester, subjects that together presented Design and Construction to students. In this case, there was the possibility of integrating students for the development of an integrated project, involving all subjects of the semester in the construction of a solution.

Relevant data extracted from the SLR were the tools used in the studies, and, consequently, in the teaching of SDC. The tools were indicated in the works; however, there was no evaluation of the impact of their usage in the teaching–learning process.

Relationships of students outside the academic environment was also an important variable for the development of soft-skills. When inserting companies and organizations within the academic context to enable experience in an organizational environment, students were able to apply the theoretical concepts learned in a corporate environment.

Based on the discussion presented in this section, we identified a proposal for teaching SDC, namely, the inclusion of content, in the teaching units/course of Software Engineering, in a practical way, having, as the main teaching strategy, the development of a project in a group, having a maximum of 6 students, with external stakeholders, and using the GitHub and UML tools. In addition, at the end of the teaching unit/course, students would be evaluated according to the final delivery of the project. This proposal was obtained due to the greater occurrence of these characteristics, in each element of the research (discipline/course, modality, teaching strategy, tool, and evaluation), among the selected studies.

Regarding the proposal presented, it can be feasibly applied, since the application of the strategy "group project development", aligned with the use of GitHub and UML tools and considering the participation of external stakeholders, provides students with practical learning on using tools in a collaborative way, which enables students to attend to a given project, with emphasis on delivering a product at the end of the teaching unit/course. In addition, the development of groups with a maximum of 6 people allows each student to experience teamwork, similar to what is experienced in the job market.

Finally, it is noted that the teaching of SDC in all its interfaces, whether in the course, in the way content is presented, in the strategies and in the form of evaluation, mainly related to the strategies and the form of evaluation.

## 6. Threats to Validity

### 6.1. Internal Validity

Internal validity specifically refers to whether or not an experimental treatment/condition makes a difference, and whether there is sufficient evidence to support the claim [29].

The process of extracting and selecting studies was carried out by the researchers following the systematic review protocol, Section 3. In addition, criteria for inclusion and exclusion of studies were defined, as well as quality criteria applied to the selected works. Detailing the review protocol is a way of mitigating the threat of internal validity, especially in the process of selecting studies.

### 6.2. External Validity

External validity refers to the generalizability of the results of the treatment/condition [30].

To mitigate risks, a review protocol was developed, detailed in Section 3, to enable the study to be replicated and the results obtained to be the same. In addition, other related SLR were identified and presented some practices similar to this study.

### 6.3. Construction Validity

Construct validity is concerned with the relationship between theory and observation [30]. The theory on teaching SDC is related to teaching Software Engineering in general and teaching Modeling and Development. In this sense, the present study obtained information that portrayed the relationship between the theory and the observation presented in the studies. Another important point was the validation of the search string, with the objective of identifying a certain work that addressed the subject in the process of extracting the works.

### 6.4. Conclusion Validity

Conclusion validity is related to the ability to reach a correct conclusion on the relationship between treatment and outcome [29].

There is a threat to this validity due to the fact that few studies presented answers to all the research questions elaborated. Therefore, the nomenclature "Not shown" was presented as the result. With a view to aligning the results and the process of extracting and selecting the studies, Section 4 presents the tabulation of the data found in a structured way in order to support the discussions and the results of the SLR.

## 7. Conclusions

The current work presented the execution of an SLR, with the objective of identifying strategies for the teaching of SDC. In all, 51 studies were selected, from a total of 302 found, through the execution of the search string in the ACM and IEEE databases, for the period from 2015 to 2022. The selection process of the studies was divided into three phases, as follows: I—Elimination by application of the established exclusion criteria, II—Elimination by Title and Abstract by application of the established inclusion criteria and III—Extraction of data and classification by Complete reading. In phase III, 11 types of information were extracted, which included the basic information of the study (title, year and authors) and the answers to the research questions, presented in Section 3.1.

As a result, the SLR identified the diversity of strategies for teaching SDC, and contemplated the most frequent active methodologies in the literature, such as Problem-Based Learning, Project-Based Learning and Flipped Classroom. However, the highlighted Formative Feedback and Athletic Approach strategies were presented as alternatives for teaching SDC. In addition to the strategies, it was identified that the teaching of SDC, for the most part, has a Practical and Theoretical–Practical teaching model, with the use of an active teaching approach.

In the process of extracting the data of the selected studies, the weight of the project in the students' final grades was observed for some works, and was presented as an element of diversity, since there were variations in the weight of projects from 33% (minimum) to 80% (maximum). Group work, often part of project development, was also highlighted among the selected studies, presenting the highest percentage. As a result, information on the number of students per group was extracted, which mostly ranged from 3 to 6 students per group.

In the discussion of the results, a proposal for teaching SDC was put forward, contemplating the most frequent result for each element of the research (teaching units/course, modality, teaching strategy, tools, and evaluation). The inclusion of content in the Software Engineering teaching unit/course in a practical way, having, as the main teaching strategy, the development of a project in a group, having a maximum of 6 students, with external stakeholders, making use of the GitHub and UML tools, and being evaluated by the final delivery of the project, presents itself as a viable alternative for teaching SDC.

Regarding future work, the review is presented as a subsidy for the proposal of curricular design for the teaching of SDC, as well as for composition of the strategies identified in a teaching methodology. In addition, exploring teaching activities that involve practices and development of work in groups, according to the result of this SLR, are presented as good practices to enhance teaching–learning.

**Author Contributions:** Conceptualization, V.d.S.C.; methodology, V.d.S.C. and S.R.B.O.; validation, V.d.S.C. and S.R.B.O.; formal analysis, V.d.S.C. and S.R.B.O.; investigation, V.d.S.C.; resources, V.d.S.C.; data curation, V.d.S.C.; writing—original draft preparation, V.d.S.C.; writing—review and editing, V.d.S.C. and S.R.B.O.; visualization, V.d.S.C. and S.R.B.O.; supervision, S.R.B.O.; project administration, S.R.B.O. All authors have read and agreed to the published version of the manuscript.

**Funding:** This research received no external funding.

**Institutional Review Board Statement:** Not applicable.

**Informed Consent Statement:** Not applicable.

**Data Availability Statement:** List of articles available at Appendix A.

**Conflicts of Interest:** The authors declare no conflict of interest.

## Appendix A. SLR Selected Studies

**Table A1.** SLR studies.

| ID | Title | Authors | Year |
|---|---|---|---|
| PS01 | Department-wide Multi-semester Community Engaged Learning Initiative to Overcome Common Barriers to Service-Learning Implementation | Timmerman, Kathleen and Goldweber, Michael | 2022 |
| PS02 | An Eco-System Approach to Project-Based Learning in Software Engineering Education | Stahl, Daniel and Sandahl, Kristian and Buffoni, Lena | 2022 |
| PS03 | Supporting Real Demands in Software Engineering with a Four Steps Project-Based Learning Approach | Silva, Leonardo Humberto and Castro, Renata Xavier and Guimaraes, Marice Costa | 2021 |
| PS04 | Improving Student Engagement with Project-Based Learning: A Case Study in Software Engineering | Morais, Paula and Ferreira, Maria Joao and Veloso, Bruno | 2021 |
| PS05 | Improvement of Learning Outcomes in Software Engineering: Active Methodologies Supported through the Virtual Campus | Garcia-Holgado, Alicia and Vazquez-Ingelmo, Andrea and Garcia-Penalvo, Francisco J. and Conde, Majose Rodriguez | 2021 |
| PS06 | Peer Instruction in Online Synchronous Software Engineering-Findings from fine-grained clicker data | Gopal, Bhuvana and Cooper, Stephen | 2021 |
| PS07 | Teaching and Learning of Interface Design: An Experience Using Project-Based Learning Approach | De Sales, Andre Barros and Boscarioli, Clodis | 2021 |
| PS08 | Students perception on the impact of their involvement in the learning process: An empirical study | Todericiu, Ioana and Serban, Camelia and Vescan, Andreea | 2021 |
| PS09 | Experience of Teaching a Course on Software Engineering Principles without a Project | McBurney, Paul W. and Murphy, Christian | 2021 |
| PS10 | A Three-Year Study on Peer Evaluation in a Software Engineering Project Course | Morales-Trujillo, Miguel Ehecatl and Galster, Matthias | 2021 |
| PS11 | SE-First: A New Approach to Software Engineering Education | Maly, Colin and Person, Suzette | 2021 |
| PS12 | Exploiting Crowd-based Learning Method in Software Engineering Course | Mao, Xinjun and Lu, Yao | 2020 |
| PS13 | Gamification applied for Software Engineering teaching-learning process | De Sousa Pinto, Fabrício and Silva, Paulo Caetano | 2017 |
| PS14 | Teaching strategies in software engineering towards industry interview preparedness | Johnson, William Gregory and Sunderraman, Raj | 2020 |
| PS15 | LEARN Board Game: A game for teaching Software Architecture created through Design Science Research | Sousa, Tamires A.S. and Marques, Anna B.S. | 2020 |
| PS16 | Towards an Evaluation Process around Active Learning based Methods | Serban, Camelia and Vescan, Andreea | 2020 |
| PS17 | Quality-driven and abstraction-oriented software construction course design: To fill the gap between programming and software engineering courses | Wang, Zhongjie and Xu, Hanchuan | 2020 |
| PS18 | Flipping Laboratory Sessions: An Experience in Computer Science | Parejo, José A. and Troya, Javier | 2020 |
| PS19 | Students perception on the use of project-based learning in software engineering education | Souza, Maurício and Moreira, Renata | 2019 |
| PS20 | Teaching software engineering tools to undergraduate students | Raibulet, Claudia and Fontana, Francesca Arcelli | 2019 |
| PS21 | Floss in software engineering education supporting the instructor in the quest for providing real experience for students | Silva, Fernanda Gomes and Tavares, Jenifer Vieira Toledo | 2019 |

**Table A1.** *Cont.*

| ID | Title | Authors | Year |
|---|---|---|---|
| PS22 | Teaching software modelling in an undergraduate introduction to software engineering | Westphal, Bernd | 2019 |
| PS23 | Pilot experience applying an active learning methodology in a software engineering classroom | Garcia-Holgado, Alicia and Garcia-Penalvo, Francisco J. and Rodriguez-Conde, Maria Jose | 2018 |
| PS24 | Design thinking and agile practices for software engineering an opportunity for innovation | Corral, Luis and Fronza, Ilenia | 2019 |
| PS25 | Software creation workshop: A capstone course for business-oriented software engineering teaching | Paiva, Sofia Costa and Carvalho, Dárlinton Barbosa Feres | 2018 |
| PS26 | Developing software engineering skills using real tools for automated grading | Heckman, Sarah and King, Jason | 2018 |
| PS27 | An Experience Report on the Use of Experience Maps and Sketches in a Database Course Project | Martínez, Alexandra | 2018 |
| PS28 | Evaluation of the university curriculum in the formation of competences for the software development industry | Enriquez, Hesmeralda Rojas | 2018 |
| PS29 | Teaching Adult Learners on Software Architecture Design Skills | Lieh, Eng Ouh | 2018 |
| PS30 | Continuous delivery of personalized assessment and feedback in agile software engineering projects | Bai, Xiaoying | 2018 |
| PS31 | Designing a reference architecture for a collaborative software production and learning environment | Restrepo Naranjo, Juan Felipe | 2018 |
| PS32 | Hiring millennial students as software engineers: A study in developing self-confidence and marketable skills | Hegee, Scott | 2018 |
| PS33 | CURRICULUM CHANGES TO IMPROVE SOFTWARE DEVELOPMENT SKILLS IN UNDERGRADUATES | O'neill, Brian | 2018 |
| PS34 | Flipping a graduate-level software engineering foundations course | Erdogmus, Hakan and Peraire, Cecile | 2017 |
| PS35 | PBL Integration into a Software Engineering Undergraduate Degree Program Curriculum: An Analysis of the Students' Perceptions | Guedes, G. T.A. | 2017 |
| PS36 | Retrospective for the Last 10 years of Teaching Software Engineering in UFC's Computer Department | De Castro Andrade, Rossana M. | 2017 |
| PS37 | Applying design thinking in disciplines of systems development | Coutinho, Emanuel F. | 2016 |
| PS38 | AGES: An Interdisciplinary Space Based on Projects for Software Engineering Learning | Yamaguti, Marcelo H. | 2017 |
| PS39 | Comparative Critiquing and Example-based Approach for Learning Client-Server Design | Jamal, Nur Amirah Amjath | 2017 |
| PS40 | Evolving a Project-Based Software Engineering Course: A Case Study | Delgado, David | 2017 |
| PS41 | Teaching Software Engineering Course with Cooperative Learning Method: A Pilot Study | Basri, Shuib | 2016 |
| PS42 | SoftBook: Software Development as an Adventure | Fernandes Silva, Lyrene | 2017 |
| PS43 | Ten years of capstone projects at Okanagan College: A retrospective analysis | Khmelevsky, Youry | 2016 |
| PS44 | STUDENT DEVELOPED COMPUTER SCIENCE EDUCATIONAL TOOLS AS SOFTWARE ENGINEERING COURSE PROJECTS | Cicirello, Vincent A | 2016 |
| PS45 | On the evaluation of student team software development projects | Tafliovich, Anya | 2015 |
| PS46 | WEB APPS IN THE COMPUTER SCIENCE CURRICULUM: A GUIDE USING HEROKU, JAVA SERVLETS, AND POSTGRES | Solheim, Jeffery | 2015 |
| PS47 | An athletic approach to software engineering education | Johnson, Philip | 2016 |
| PS48 | Reflections on applying constructive alignment with formative feedback for teaching introductory programming and software architecture | Cain, Andrew | 2016 |
| PS49 | Assessing problem-based learning in a software engineering curriculum using Bloom's Taxonomy and the IEEE software engineering body of knowledge | Dolog, Peter | 2016 |
| PS50 | Teaching Software Architecture to Undergraduate Students: An Experience Report | Rupakheti, Chandan R. | 2015 |
| PS51 | Drawing Insight from Student Perceptions of Reflective Design Learning | Wilkins, Thomas V. | 2015 |

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
