# Peer review of "Diversity in Software Design and Construction Teaching: A Systematic Literature Review"

_education, doi:10.3390/educsci13030303_

Round 1
Reviewer 1 Report
In the attached file

Reviewer 2 Report
Unfortunately, in this form, there are too many figures missing.
Maybe the entire list of articles (as of SLR subject) could be included in the article, and not as an external file link.
Q1 response argues the teaching units approached, not quite the "pedagogical projects".
Q2 response mentions an interesting argument related to active/passive approaches.
Q3 response is probably the most valuable due to the identification of valid teaching strategies. Concerning the tools, maybe only the first 10 of them are worth mentioning.
Q4 maybe is not quite well/clearly formulated. Also, perhaps only a top 10 of strategies could be enough. Table 9 (project weight) is a valuable contribution.
Discussion and conclusions are fair and reflect the contributions in the field.
Round 2
Reviewer 1 Report
The authors have considered all the clarifications provided by me and the other reviewer. The article is clearer and more coherent now.
Please see two additional comments, which have to be changed:
1. Figures: Figures 6,9 show a relevant and accepted pie, while figures 3,8 also represent pies arranged in a square: (*) Why the difference? There is no reason why there should not be uniformity when it comes to presenting parts of a whole (100%); It is not clear to me personally why the change and presentation in a square is necessary, a round pie is accepted, and the reader should not look for a reason for a different graphical representation; (*) There is no reason in my opinion for showing the graphs 4,5,7 which do not constitute a whole, and the illustration does not contribute anything to understanding; the percentages can be simply written in continuous text, or if the emphasis is important in the table; The illustration does not contribute anything.
2. The appendix should appear after the references.
